# Bacterial Communities in the Embryo of Maize Landraces: Relation with Susceptibility to Fusarium Ear Rot

**DOI:** 10.3390/microorganisms9112388

**Published:** 2021-11-19

**Authors:** Alessandro Passera, Alessia Follador, Stefano Morandi, Niccolò Miotti, Martina Ghidoli, Giovanni Venturini, Fabio Quaglino, Milena Brasca, Paola Casati, Roberto Pilu, Davide Bulgarelli

**Affiliations:** 1Department of Agricultural and Environmental Sciences–Production, Landscape, Agroenergy, Università degli Studi di Milano, Via Celoria 2, 20133 Milan, Italy; alessia.follador@unimi.it (A.F.); niccolo.miotti@unimi.it (N.M.); martina.ghidoli@unimi.it (M.G.); gventurini@isagro.com (G.V.); fabio.quaglino@unimi.it (F.Q.); paola.casati@unimi.it (P.C.); salvatore.pilu@unimi.it (R.P.); 2Institute of Sciences of Food Production, Italian National Research Council, Via Celoria 2, 20133 Milan, Italy; stefano.morandi@ispa.cnr.it (S.M.); milena.brasca@ispa.cnr.it (M.B.); 3Plant Sciences, School of Life Sciences, University of Dundee, Invergowrie DD2 5DA, UK; d.bulgarelli@dundee.ac.uk

**Keywords:** *Fusarium verticillioides*, 16S metabarcoding, digital PCR, RAPD, Firmicutes

## Abstract

Locally adapted maize accessions (landraces) represent an untapped resource of nutritional and resistance traits for breeding, including the shaping of distinct microbiota. Our study focused on five different maize landraces and a reference commercial hybrid, showing different susceptibility to fusarium ear rot, and whether this trait could be related to particular compositions of the bacterial microbiota in the embryo, using different approaches. Our cultivation-independent approach utilized the metabarcoding of a portion of the 16S rRNA gene to study bacterial populations in these samples. Multivariate statistical analyses indicated that the microbiota of the embryos of the accessions grouped in two different clusters: one comprising three landraces and the hybrid, one including the remaining two landraces, which showed a lower susceptibility to fusarium ear rot in field. The main discriminant between these clusters was the frequency of Firmicutes, higher in the second cluster, and this abundance was confirmed by quantification through digital PCR. The cultivation-dependent approach allowed the isolation of 70 bacterial strains, mostly Firmicutes. In vivo assays allowed the identification of five candidate biocontrol strains against fusarium ear rot. Our data revealed novel insights into the role of the maize embryo microbiota and set the stage for further studies aimed at integrating this knowledge into plant breeding programs.

## 1. Introduction

Maize (*Zea mays* L.) is one of the most widely grown crops worldwide. It is not only a staple food for people in several countries, but also a very important crop for animal feed and finds use as industrial material for production of fuels, among other technological uses. In addition to all these practical applications, maize has also been used as a model organism in scientific research, as it benefits from a high level of phenotypic and genetic variability, and quick life cycle. 

While these characteristics allowed for a great increase in yield in the past century [1], the predicted demands for food in the next few decades [2] suggest that substantial changes in breeding techniques and agronomic processes will be required to reach the needed level of crop improvement and yield [3].

The history of maize as a crop is very long: its domestication is traced back to approximately 8700 BC in central America and the Spanish brought it to Europe at the end of the 16th century, where favorable environmental and social conditions allowed it to be employed with success [4]. In particular, the first written reports on the use of corn in Italy date back to 1600, proving that maize had well adapted to the climatic zone of cultivation and to the local tradition of the people living in the North-Eastern part of the peninsula. The cultivation in Italy gave rise to several landraces which were later abandoned for the more productive dent hybrids when mechanized farming practices became more common in the second half of the twentieth century [4].

The hybrids, which were selected mostly for their massive yield traits in field, have lost several useful features which were present in ancient maize varieties that gave the landraces a particular biochemical composition, including relevant nutritional components such as antioxidants and carotenoids [5], and pathogen resistance. It has been recently reported that certain Italian maize germplasm can be advantageous in resisting mycotoxin accumulation in its kernels [6,7]. This last trait is of particular importance not only for the ability of the maize plants to produce an abundant quantity of kernels, but also for their quality: several phytopathogenic fungal species belonging to the genera *Aspergillus*, *Fusarium*, and *Penicillium* that frequently colonize maize plants in field are known to produce mycotoxins as part of their secondary metabolism [8]. These fungi may not always produce visible damage to kernels but may still contaminate them with mycotoxins. These toxins, which are highly stable and extremely dangerous even at low concentrations, are currently one of the main concerns for human and animal nutrition all over the world [9]. The use of endophytes with biocontrol abilities against these toxigenic fungi can be of great interest for several reasons: biocontrol endophytes may prove effective against these fungi that develop at least for a portion of their life cycle inside the tissues of the host, and some endophytes are known to chelate and detoxify mycotoxins [10,11]; furthermore, the use of synthetic fungicides has given inconsistent results on this particular issue, sometimes resulting in a greater rate of mycotoxin production by the pathogens [12], making the use of biological control agents a more sustainable and potentially effective option [13].

While the local varieties have disappeared from the fields, many efforts were made to recover and preserve the genotypes of these traditional populations. These landraces are now receiving much attention for their potential use in new breeding programs with the aim of identifying novel alleles and haplotypes and using them in a context of low-input and sustainable agriculture [14,15,16,17]. 

Despite this, alleles and haplotypes are only a part of what should be taken into account while carrying out breeding programs: quantitative trait loci and, in general, genome-wide association studies of complex physiological traits consistently found that associated genetic factors, such as allelic polymorphisms or DNA mutations, only explained a minority of the expected heritable fraction, highlighting the presence of other factors contributing to the variability of these traits [18]. This discrepancy is known as “missing heritability”, and the microbiome could be one contributor to this heritability that is not explained by the genome sequence alone.

While it has been held for a long time that healthy plant tissues were sterile, the presence of complex communities of microorganisms inside every plant tissue has been proven [19,20]. These microorganisms, called endophytes, can have beneficial, neutral, or harmful effects on the host. Since beneficial endophytes can influence the growth of the host plant, as well as its metabolic processes and resistance to both biotic and abiotic stresses [21], their presence can greatly affect the phenotype of their host. By increasing the uptake of nutrients and granting higher resistance to pathogens and pests, as well as other stresses, a positive relation between microorganisms, the plant, and the environment can greatly contribute to integrity, proper functionality, and sustainability of agro-ecosystems [22]. 

Moreover, seed-borne endophytes have been shown to be an important source of bacteria within other tissues. The identification of a set of endophytic microbes among *Zea* spp. that are conserved across evolutionary and ecological boundaries [19] suggests microbes with beneficial properties to the host plant are selected from the environment by the plants themselves. In addition to environmental origin there is also evidence that, in some plant species, bacterial endophytes can be inherited from one generation to the next through seed [23,24].

The aim of this study was to characterize, using both cultivation-dependent and -independent approaches, the embryo-associated bacterial microbiota of six maize accessions, including a hybrid and five landraces with distinct geographical origins, in order to determine whether this microbiota and/or single bacteria that compose it can have an effect in contrasting the infection from *Fusarium verticillioides*, a main agent of fusarium ear rot in Northern Italy.

## 2. Materials and Methods

### 2.1. Plant Material

This study was carried out on seeds obtained from six maize accessions, all cultivated at the experimental field of the University of Milan situated in the Angelo Menozzi farm in Landriano (PV), Italy (45°190 N, 9°160 E, 88 m a.s.l). The six accessions, which include a hybrid B73 × Mo17 and five landraces with different characteristics, are reported in Table 1. All accessions were sowed in the first half of May in years 2017, 2018, and 2020, and harvested in the last week of August. The seeds from years 2017 and 2018 were dried and stored at room temperature for further analyses. The material from year 2020 was used exclusively for the infection assay and the whole ears were analyzed right after harvest.

### 2.2. Description of Bacterial Embryo-Associated Community

The bacterial community associated to the embryo of the selected maize accessions was described with both cultivation-dependent and cultivation-independent methods, detailed as follows:

#### 2.2.1. Sterilization of Seeds and Embryo Isolation

Isolation of cultivable bacteria was carried out from the embryo of maize seeds of the six accessions collected in year 2017. Isolation starting from different matrixes (whole seeds or endosperm) were considered and tried out on some of the samples but, ultimately, extraction from embryos was deemed more effective (Appendix A).

Seeds were surface sterilized by a wash in ethanol 70%, followed by a wash in bleach 2.5%, and another wash in ethanol 70%. Each wash had a duration of 3 min. After the wash steps, the seeds were rinsed in sterile water for three times. An aliquot of the last washing water was plated on lysogeny broth agar (LBA) plates and incubated at 24 °C to check for growth and confirm the efficacy of the sterilization.

The sterilized seeds were kept in sterile water at 20 °C in the dark for 24 h to soften them, and then the embryos were excided using sterile tweezers and scalpels.

#### 2.2.2. Isolation of Cultivable Bacteria

For each of the six accessions, three samples made of ten embryos each were ground in a mortar containing 2 mL of sterile Ringer solution (Sigma-Aldrich, St. Louis, MO, USA), obtaining approximately 1:10 dilution (*w/v*). The homogenized solution of ground maize embryos was serially diluted twice, for a total of three different concentrations of 1:10, 1:100, and 1:1000.

From each of these solutions, a 100 µL aliquot was inoculated on three LBA and three tryptic soy agar (TSA) plates and incubated at 24 °C for 14 days, checking for bacterial growth every two days.

Bacterial colonies were isolated based on the phenotype of the colony: for each accession and growth medium, only one colony with a specific morphology was isolated, while colonies with the same phenotype but isolated from different maize accessions or on different medium were isolated separately.

All isolated colonies were maintained on separate plates containing the same medium as the original plate they were isolated from (LBA or TSA). Isolates obtained were given an identifier that includes the code of the accession from which they were isolated and a progressive number.

#### 2.2.3. Molecular Characterization of Cultivable Bacteria

From each isolate, DNA was extracted following the protocol described by Wilson [27]. Briefly, this method starts from an overnight culture in liquid broth of the bacterial strain, and extracts the nucleic acids by lysis with lysozyme, incubation with protease K and SDS, and incubation with a CTAB buffer, separation with chloroform:isoamyl alcohol, washing with ethanol, and lastly suspension of the nucleic acids in TE. The quality, quantity and integrity of the DNA was assessed with a NanoDrop ND-1000 spectrophotometer (Thermo Fisher Scientific, Waltham, MA, USA) and by electrophoresis on 1% agarose gel.

A first step in characterizing the isolates was carried out through a RAPD-PCR approach, following the protocol described by Morandi and colleagues [28], utilizing a single primer (M13: 5′-GAGGGTGGCGGTCT-3′) to obtain amplicons of different lengths, which pattern can be used in taxonomic fingerprinting. 

The amplicons were visualized through electrophoresis in a 1.5% agarose gel in TAE, using SYBR Safe (Invitrogen, Waltham, MA, USA) dye. The obtained amplification profiles for each isolate were grouped together through an UPGMA algorithm using the BioNumeric 5.0 package (Applied Maths, Sint-Martens-Latem, Belgium). Different isolates that showed more than 90% identity in the RAPD-PCR profile, came from the same maize accession and showed identical morphology were considered to be the same isolate for subsequent characterization steps, and only one representative isolate was utilized from each group.

From each of these representative isolates, an approximately 1400 bp portion of the 16S rDNA gene was amplified by PCR using the 27F/1492R primer pair (27F: 5′-AGAGTTTGATCMTGGCTCAG-3′; 1492R: 5′-ACCTTGTTACGACTT-3′ [29]). The PCR mix contained 1× GoTaq Flexi buffer (Promega, Madison, WI, USA), 1.5 mM MgCl_2_, 0.5 µM of each primer, 200 µM dNTPs. 2.5 U of Taq DNA polymerase, 2 µL of template DNA, and water up to 50 µL. Amplification was carried out with an initial denaturation at 94 °C for 5 min, 35 cycles of denaturation at 94 °C for 1 min, annealing at 53 °C for 1 min, and extension at 72 °C for 1.5 min, followed by a final extension step at 72 °C for 7 min. The amplicons were visualized through electrophoresis in a 1% agarose gel in TBE, using Midori Green (Nippon Genetics, Düren, Germany) dye. The obtained amplicons were sequenced in both senses (5× coverage per base position) by a commercial service (Eurofins Genomics, Ebersberg, Germany). Nucleotide sequences were compiled in FASTA format, assembled by employing the Contig Assembling Program of the software BioEdit version 7.2 [30]. Assembled sequences are available on NCBI under accession numbers from OK584315 to OK584384. Attribution of each isolate to species or genus was carried out by comparing the 16S sequence obtained with those present in NCBI database using the MegaBLAST algorithm.

#### 2.2.4. Cultivation-Independent Description of Bacterial Community

For each maize accession, three samples containing ten embryos each were prepared starting from seeds obtained in year 2017 and in year 2018. From these samples, DNA was extracted following a CTAB-based protocol described by Angelini and colleagues [31]. Briefly, this protocol allows the extraction of DNA from plant material by grinding the plant tissue in a CTAB buffer, followed by incubation at high temperature, separation with chloroform:isoamyl alcohol, precipitation with isopropanol, washing with ethanol, and resuspension of the nucleic acids in TE.

The quality, quantity and integrity of the DNA was assessed with a Nanodrop1000 spectrophotometer and by electrophoresis on 1% agarose gel.

On these samples, the hypervariable V4 region of the 16S rRNA gene was amplified using the primer pair 515F/806R (515F: 5′-GTGCCAGCMGCCGCGGTAA-3′; 806R: 5′-GGACTACHVGGGTWTCTAAT-3′). These primers contained an Illumina flow cell adapter at their 5′ end and the reverse primer also included a 12 bp unique barcode sequence to allow simultaneous sequencing of multiple samples. In this reaction, blocking primers with a specific sequence for chloroplasts/mitochondria were added to reduce the quantity of plant derived sequences obtained as detailed by Moronta-Barrios and colleagues [32].

After PCR, the obtained amplicons were pooled and submitted for an Illumina MiSeq sequencing, 2 × 150 bp chemistry, to the Genome Technology group at the James Hutton Institute (Invergowrie, UK). Quality control, processing, and sequencing were carried out as previously described [33,34,35].

Sequencing reads were analyzed through a custom bioinformatics pipeline. The first step was carried out in the QIIME software, version 1.9.0, to process the FASTQ file, using default parameters for each step [36]. Using the join_paired_ends.py command, forward and reverse files for each library were decompressed and merged, using a threshold of 30 bp overlap between reads. After this process, reads were demultiplexed according to their initial barcode sequences. Using the split_libraries_fastq.py command, the reads were filtered for quality, using a threshold on the PHRED score ‘-q’ of 20. These high-quality reads were trimmed at a uniform length of 250 nucleotide through the ‘fastq_filter’ function of USEARCH [37]. These truncated sequences were then used for clustering in Operational Taxonomical Units (OTUs), using the threshold value of 97% identity value. After clustering, the identity of the OTUs was determined with a closed reference OTU-picking approach against the Silva database (version 132) [38] using the SortMeRNA algorithm [39]. The output of this procedure was an OTU table reporting the abundance of each OTU for each sample, as well as a phylogenetic tree. Singletons, defined as OTUs found only once in the whole dataset, and plant derived sequences, those attributed to chloroplasts and mitochondria, were removed from the dataset with the filter_otus_from_otu_table.py command. 

The OTU table and phylogenetic tree were used as input files in R [40] to be analyzed with the following packages: Phyloseq [41] for processing and calculating metrics of Alpha- and Beta-diversity; Vegan [42] for statistical analysis of Beta-diversity; Ape [43] for phylogenetic tree analysis; ggplot2 [44] for data visualizations.

Alpha-diversity was calculated using the estimate richness function in Phyloseq, and expressed through the Observed, Chao-1, and Shannon indexes.

Beta-diversity was analyzed on a normalized table, in which the uneven number of reads per sample was normalized converting the individual OTU count to an abundance percentage, using the Weighted Unifrac metric, which is sensitive both to relative abundance and phylogenetic classification, and the Unweighted Unifrac metric, which is sensitive to unique taxa [45]. The dissimilarity matrix obtained from this process was visualized using the ordinate function in the Phyloseq package and its significance was assessed through a permutational ANOVA (5000 permutations). The effect size and statistical significance of variables in the Beta-diversity dissimilarity matrix was inspected by a permutational multivariate analysis of variance (PermANOVA) using the Adonis function in the Vegan package (10,000 permutations).

The sequencing reads utilized in this analysis are available at ENA PRJEB47936, while the files and script–OTU table, mapping files, R script–are available on GitHub (https://github.com/AlessandroPasser/Maize_Embryo_Microbiota (accessed on 16 November 2021)).

#### 2.2.5. Validation of Sequencing Data

The results obtained from the sequencing of 16S were validated through digital PCR. In particular, total quantity of bacteria and Firmicutes in each DNA sample extracted from maize embryos in 2018 was evaluated using the primer pairs 906F/1062R (906F: 5′- AAACTCAAAKGAATTGACGG-3′; 1062R: 5′-CTCACRRCACGAGCTGAC-3′) and 928F-Firm/1040R-Firm (928F-Firm: 5′-TACGGCCGCAAGGCTA-3′; 1040R-Firm: 5′-TCRTCCCCACCTTCCTCCG-3′) [46], respectively. The amplification was carried out using an the EVAGREEN MIX (Qiagen, Hilden, Germany), following the manufacturer’s instructions, using a final primer concentration of 300 nm and loading 2 µL of each DNA sample at 5 different concentrations, ranging from 1:10 dilution to 1:104. Controls included in the reaction include two bacterial DNAs extracted from pure culture: DNA from a *Bacillus pumilus* strain to act as positive control in both reactions, and DNA from a *Pseudomonas syringae* strain to act as positive control with the universal bacterial primers and as negative control in the Firmicutes-specific primers. No-Template Controls were included, adding sterile water instead of DNA to the mix. All reactions were carried out on a QIAcuity machine, using 96-wells plates with 8500 partitions per well, and data were analyzed with QIAcuity Suite v 1.3.

For each sample, the count of copies/µL was converted to copies/ng of DNA in the original sample to normalize the data. The copy number of total bacteria and Firmicutes was expressed as an average of all analyzed samples for each maize accession and the ratio between the two was compared between the data obtained from dPCR and from Illumina sequencing.

### 2.3. In Vitro Characterization of the Antifungal Properties of the Isolated Bacteria

The bacteria isolated from the maize accessions were evaluated through different in vitro assays to determine whether they had some antifungal activity towards a plant pathogenic fungus widely involved in fusarium rot in northern Italy, *Fusarium verticillioides*.

#### 2.3.1. In Vitro Antifungal Assays

The ability of the bacterial strains to reduce the growth of *F. verticillioides* strain GV2245 (identified as FV from now on), isolated from maize ear in 2011, was assessed in a two-steps experiment utilizing dual-culture technique.

In the first step, a qualitative assay was set up, streaking a single colony of the bacterial isolate on one side of the plate, containing tryptone glucose yeast extract agar medium (TGYA; containing 5 g/L tryptone, 1 g/L glucose, 3 g/L yeast extract, 15 g/L agar). After two days of incubation at 24 °C, FV was inoculated as a mycelium/agar plug (0.6 cm in diameter) on the diametrically opposite side of the plate, and it was incubated at 24 °C for seven days. Each plate was produced in triplicates, and control plates in which FV alone was growing were prepared as well. In each plate the inhibition of the growth of FV was deemed successful if an inhibition halo was visible in the plate, and unsuccessful if no halo was seen (Appendix A). At the end of this period, the interaction between bacterial strain and fungus was classified as follows: a number from 0 to 3 was assigned to each bacterial isolate, based on the number of replicates in which the bacteria managed to halt the growth of FV. Bacteria that displayed an antifungal activity in most of the replicates (class 2 and 3) were considered for the second step of the experiment.

The second step, a quantitative assay, was carried out as described by Passera and colleagues [47]. Briefly, 20 µL droplets from overnight liquid culture of each strain were placed on four cellulose disks near the border of a Petri dish containing TGYA and, after two days of incubation, a FV plug was placed in the middle of the plate. As negative control, plates containing FV alone were used.

Fungal growth was measured 3- and 7-days post inoculation (dpi) as mycelial growth diameter. Each test was carried out with plates in triplicate and three independent measures were made for each plate at each measuring time. Growth inhibition percentage (GIP) was calculated as [1 − (D1/D2)] × 100, where D1 is the radial colony growth on bacteria-treated plate, D2 is the radial colony growth in the control plate. Examples of plates with different growth patterns of FV in this assay are provided in Appendix A.

#### 2.3.2. In Vivo Biocontrol Assay

The most promising bacterial isolates that do not belong to potentially harmful species (i.e., species that are reported in literature as human pathogens and/or food contaminants) were chosen to carry out biocontrol assays against *F. verticillioides* on maize kernels.

Maize kernels from the hybrid accession were sterilized as previously described in Section 2.2.1 and stored overnight at 4 °C in dry, sterile tubes.

The next day, they were inoculated with a bacterial suspension of one of the selected isolates (20 mL of approximately 10^5^ CFU/mL) and incubated with agitation (150 rpm) at room temperature for three hours. After this time, a conidia suspension obtained from a mix of seven *F. verticillioides* strains (identified as FVm from now on) was added to the tube (final concentration of 10^4^ conidia/mL) and incubated with agitation at room temperature for three hours. This mixture of strains was used to avoid specific incompatibility between a single *F. verticillioides* strain and the utilized maize accession, and comprises the following *F. verticillioides* strains: Fv2003, Fv2010, Fv2170, Fv2198, Fv2221, Fv2232, and GV2245 of the collection of the University of Milan Department of Agricultural and Environmental Sciences (DiSAA, Milan, Italy). After the incubation, the maize kernels were placed in a Petri dish containing 1% agar-water medium. Ten kernels were placed in each plate, and five replicates were made for each treatment. In addition to the treatments, the following controls were made: (i) kernels inoculated only with water, in place of both bacteria and FVm, identified as C-; (ii) kernels inoculated with water in place of bacteria, but inoculated with FVm, identified as C+; (iii) kernels treated with beneficial bacterial strains not isolated from maize and previously reported to have no in vitro antifungal effect towards FV: identified as *Paenibacillus pasadenensis* strain R16 [47], *Pseudomonas syringae* strain 260-02 [48], and *Lysinibacillus fusiformis* strain S4C11 [49]. These last controls were used to determine whether the biocontrol ability displayed in this assay could be determined by specific antifungal abilities of the strains or could just be due to an aspecific interaction caused by the presence of a high concentration of bacterial cells in the suspension.

The plates were stored at 24 °C for one week and were then observed to determine the severity of the symptom caused by FVm on the germinated kernels. Each maize seedling was attributed to a class based on the observed symptom, as follows: 0—no visible symptom and no macroscopic mycelium observed on the surface; 1—no visible symptom, but presence of some macroscopic mycelium structures on the seedling surface; 2—parts of the seedling, either root or shoot but not both, are showing damage caused by FVm and a big part of the seedling is covered in mycelium; 3—the seedling is completely covered in mycelium and died, or the seed did not germinate and shows abundant mycelium on the surface. Visual examples of each symptom class are provided in Appendix A.

The data obtained in this way was converted to an infection degree percentage (ID%) for each plate, using the formula presented by Townsend and Heuberger [50].

### 2.4. In Field Susceptibility Assay of Selected Maize Accessions

In the end of July 2020, approximately seven days after the flowering of maize, when the silks were still green but started to dry from the tips, maize plants were either experimentally inoculated with FVm or mock inoculated with sterile water. The inoculation was performed by injecting 1 mL of FVm or sterile water in the silk channel of the primary ear of each plant. For each accession, five plants were inoculated and five plants were mock inoculated [9]. After harvesting the ears of maize, they were visually observed to determine the damage caused by fusarium rot on each ear. The damage was expressed as percentage of ear affected by the pathogen.

### 2.5. Statistical Analyses

Severity of FV-induced symptoms obtained during in vivo biocontrol assays (2.3.2) and in field susceptibility assays (2.4) were compared between treatments by one-way ANOVA followed by Tukey’s post-hoc test (*p* < 0.05) carried out in SPSS statistics v27 (IBM, Armonk, NY, USA).

## 3. Results

### 3.1. Gram-Positive Bacteria Dominate the Culturable Fraction of the Maize Embryo Bacterial Microbiota

The isolation and cultivation of bacterial colonies from the embryo of different maize accessions allowed us to identify 18 to 82 colonies from each accession, for a total of 303 distinct isolates (Table 2). After cultivation in axenic conditions and a further morphological assessment we reduced the number of unique bacterial isolates to 82. Next, we used RAPD-PCR profiles (Appendix A) to identify isolates sharing genomic content and this analysis further reduced the number of isolates to 70 (Table 2). The sequencing and characterization of the 16S rRNA gene of these unique isolates revealed that the collection is represented by 24 species assigned to 11 genera (Appendix A). Of note, the great majority of these bacteria (66 out of 70) was assigned to gram-positive genera: *Bacillus* (29 isolates), *Brevibacillus* (1), *Brevibacterium* (1), *Kocuria* (1), *Lysinibacillus* (5), *Microbacterium* (1), *Micrococcus* (18), *Rothia* (1), and *Staphylococcus* (9). The remaining four isolates belong to the Alpha- and Gammaproteobacteria classes, represented by genera *Brevundimonas* (1) and *Pseudomonas* (3), respectively. 

Among these isolates, some were characterized as potential food contaminants belonging to *Bacillus cereus* group (1) and to *Micrococcus luteus* species (12), and therefore not retained for further characterization.

### 3.2. The Host Genotype as a Driver of the Maize Embryo Bacterial Microbiota

To gain further insights into the determinants of the maize embryo bacterial microbiota, we performed a 16S rRNA gene amplicon sequencing survey of the six maize accessions sampled in two different years. The sequencing was done three times, once without using any sort of blocking techniques for organellar DNA, once using PNAs to interrupt elongation of sequences belonging to organelles as described by Lundberg and colleagues [51], and the final approach using blocking primers that selectively hinder the production of sequences belonging to organelles by competitive binding, as described in Materials and Methods (Section 2.2.4) [32]. The final approach used for the analysis yielded, after filtering in silico sequencing reads of low quality and mapping plant-derived sequences (i.e., mitochondria and chloroplasts), a total of 6007 reads assigned to bacteria, much higher than the 2540 obtained employing the PNA blocking approach, and comparable to the 6077 obtained using no blocking techniques. The initial number of reads obtained was 2,617,250. After having identified sequencing reads assigned to chloroplast, mitochondria and putative contaminants previously identified in the lab [35], more than 99% had of these reads had to be discarded, yielding an average of 500 reads per accession per year (min 53 reads, max 1163 reads) (Table 3). These reads were assigned to a number of OTUs per accession that ranges from 23 to 166 (Table 3).

The number of shared OTUs between samples of the same accession in different years, that might constitute a core microbiota for that accession, is represented by UpSet graphs (Figure 1). As these graphs show, only a small number of OTUs were conserved between the different years investigated: these core bacteria could be as low as 8% of the total number of OTUs (for accession C), to as high as 28% (for accession H). Considering instead all the accessions in any given year, three and eight OTUs were common among all accessions in 2017 (Figure 1G) and 2018 (Figure 1H), respectively.

Analysis of Alpha diversity highlighted very close values for the number of observed OTUs and Chao-1 index (Figure 2A), suggesting that despite the low number of total reads assigned to bacteria, the depth of sequencing might be adequate. There were no statistically significant differences between sampling years or accessions on the values of Observed OTUs (*p*-value = 0.166), Chao-1 index (*p*-value = 0.182), or Shannon’s index (*p*-value = 0.547), according to a One-Way ANOVA test.

Next, we investigated whether phylogenetic footprint could be identified within the maize embryo bacterial microbiota in the two examined years. To accomplish this task, we computed a Weighted Unifrac distance, which takes into account both the OTUs abundance and taxonomic affiliation. This analysis revealed that microbiota composition in maize embryos can be partitioned in two main groups of variety separated along the axis accounting for ~23% of variation of principal component analysis (Figure 2B). Interestingly, these groups emerged as sufficiently robust to withstand the year-to-year variation observed for the number of reads and OTUs detected. The only exception to this is accession W, which shows a variable attribution to these groups in different years, clustering with accession N in 2017 and with the other four accessions in 2018. Congruently, a permutational analysis of variance indicated the maize accession as a significant determinant of the maize microbiota (Adonis, R^2^ = 0.315, *p* value = 0.000, 10,000 permutations) while neither the year (Adonis, R^2^ = 0.034, *p* value = 0.359, 10,000 permutations) nor the interaction between year and accession (Adonis, R^2^ = 0.167, *p* value = 0.388, 10,000 permutations) emerged as significant factors driving microbiota composition. Strikingly similar results were obtained when we computed unweighted Unifrac distances, which are sensitive to unique taxa, regarding the relevance of genotype (Adonis, R^2^ = 0.238, *p* value = 0.003, 10,000 permutations). This indicates that compositional changes identified in our survey are sufficiently robust to withstand biases introduced by the sequencing protocol.

All accessions, with the exception of N in 2017, had a prevalence of Proteobacteria in their microbiota, constituting even more than 50% of all the bacterial reads (Figure 3A). All other phyla have great differences in abundance between the different accessions. In particular, the main differences between N (both years), W (2017) accessions and the other 4 (both years) are: higher abundance of Firmicutes, low abundance of Bacteroidetes, Chloroflexi, Planctomycetes, and Verrucomicrobia.

At family level, the situation reflects the one detected at Phylum level. The presence of Proteobacteria is, for all accessions except W in 2017, due mostly to bacteria belonging to the Burkholderiaceae family, and to a minor extent from Pseudomonadaceae and Xanthomonadaceae (Figure 3B). For the accession W in 2017 the majority of Proteobacteria reads belong to the Enterobacteriaceae family instead. Regarding the Firmicutes, which are the phylum with higher relative abundance in accessions N (both years) and W (2017) compared to the others, the great majority of the reads belong to the Bacillaceae family.

Considering the relatively low number of obtained reads, a complementary approach was employed to ascertain the quantitative nature of our results. As the abundance of Firmicutes emerged as a distinctive feature of the maize embryo microbiota across accessions, we employed a quantitative digital PCR to assess the abundance of members of this phylum as a proportion of 16S rRNA gene copy number. This investigation revealed a correlation of R^2^ = 0.944 (Figure 4, *p*-value = 0.000) between the two quantification approaches, suggesting that, despite the limited number of amplicon sequencing reads, the Firmicutes-led microbiota diversification represents a genuine trait of the communities inhabiting maize embryos.

### 3.3. Gram-Positive Bacteria Isolated from Maize Show Strong Antifungal Ability against F. verticillioides

The bacterial strains isolated from maize embryos were screened for their ability to inhibit the growth of *F. verticillioides* in vitro and on maize kernels. The in vitro assay followed two steps of screening, a first, qualitative one that identified 29 strains that were able to develop an inhibition halo and stop the mycelial growth of *F. verticillioides* (Appendix A). These strains were isolated from the different accessions as follows: no strains from accession A, two from accession C, one from accession G, 17 from accession H, five from accession N, and four from accession W. These 29 strains were used in a subsequent assay that quantified the amount of inhibition the bacteria could exert. Of these strains, 16 showed a growth inhibition percentage (GIP) above 20% at 7 dpi, which was considered as a threshold to be considered for further trials (Figure 5A). From these strains, six were selected for the in vivo assay with maize kernels: the strains were selected by considering the taxonomic attribution (ignoring strains that belong to species known as possibly pathogenic), the GIP produced, and their ability to grow rapidly and consistently in laboratory. From these characteristics, the six chosen strains were G01 (*Pseudomonas azotoformans*), H03 (*Micrococcus yunnanensis*), H07 (*Kocuria rosea*), H20 (*Bacillus megaterium*), N02 (*Bacillus paralicheniformis*), and N09 (*Bacillus paralicheniformis*). Three of these bacteria belong to the Firmicutes phylum, and two were isolated from the accession N which showed a different microbiota structure compared to the other accessions. The results of the inoculation of the conidia *F. verticillioides* strains and bacterial solutions on maize kernels are reported in Figure 5B. The positive control, inoculated only with FVm and no bacterial strain, displayed high values of ID% (up to 68%) while the negative control, mock-inoculated with water, gave lower ID% values, usually below 20%. The negative control does not yield 0% ID% as the maize kernels utilized show presence of endogenous microorganisms causing rot, apparently different *Fusarium* strains based on the symptoms caused and the appearance of the mycelium, that can lead to infection even in the absence of external inoculum. All the treatments with the different selected strains isolated from maize, except for strain H03, were associated with a significant reduction of ID% that reached values comparable to those of the negative control (Figure 5B, one-way ANOVA, Tukey’s post-hoc, *p*-value = 0.000). This result suggests that the application of the bacterial strains on the surface of the maize kernels managed to negate the effect of the exogenous FVm application but did not eliminate the ability of the naturally present, endogenous microorganisms to rot the seeds. On the other hand, the bacterial strains not isolated from maize and that did not show antifungal activity against *F. verticillioides* strains in previous studies had no effect on the ID%, which remained on values comparable to those obtained in the positive control.

### 3.4. Accessions Characterized by Similar Embryo Microbiota Show Similar Susceptibility to Fusarium Ear Rot

The susceptibility to the disease associated with *F. verticillioides*, fusarium ear rot, was assayed in field, using both non-inoculated maize ears (natural infection) and performing an experimental inoculum of the silk channels with a suspension of conidia from *F. verticillioides* strains. The severity of the disease was assessed visually, as percentage of the ear that was infected with *Fusarium* (Figure 6). All the maize accessions gave similar infection rates under natural infection: while accession N and W had lower infection than the other 4, even though this difference was not statistically significant (Figure 6A). Conversely, in the *F. verticillioides* inoculated ears, statistically significant differences were reported: accessions N and W had infection values similar to the naturally infected ears, while the remaining four accessions had way higher infection values. This can also be seen by the pictures of representative ears of maize of the different accessions (Figure 6B–G). This suggests that these two accessions may have an adaptive advantage under conditions conducive to the establishment of *Fusarium* infections.

## 4. Discussion

Finding new and effective solutions to the problem of mycotoxins in maize has been a primary challenge for sustainable agriculture since decades [8], as their presence in the kernels affects not only the direct consumption of corn as food but also animal feed, with effects on the meat and dairy production lines. For this reason, development of fungi-resistant maize genotypes could be a revolutionary advance in agriculture. Sadly, the genetic bases behind resistance to *Fusarium* spp. is not well-characterized, making breeding programs aimed in this direction very hard. Recent evidence suggests that other phenotypes which are not easily explained by simple genetic traits, such as the phenomenon of heterosis, might be related to a contribution of the microbiota [52]. A great involvement of the microbiota is expected in processes based in plant-microbe interactions such as defense mechanisms to pathogens, in which the recruitment of symbionts that can protect against pathogens is already reported [53]. This involvement is even more relevant while the plants are at the seed stage, during which they are exposed to several biotic stresses and are at their most vulnerable [54].

The situation regarding infection by *F. verticillioides* is further complicated by the different mechanisms by which it can infect maize. Infection can occur both by interaction with the silk channels [55], the mechanism that was investigated in the present study, and by entering through holes and wounds present on the surface of the plant, such as those caused by insects, in particular *Ostrinia nubilalis*. Also, *F. verticillioides* can survive as an endophyte [10], causing no symptoms to the host plant, for long periods of time and therefore give rise to latent infections that only show symptoms at later times. It is also known that different fungal pathogens, such as different species of *Fusarium* or *Fusarium* with *Ustilago* species, can interact between themselves to promote or hinder the development of disease [55].

With these considerations in mind, both regarding the complexity of the pathosystem and the involvement of microorganisms in the infection process, the hypothesis behind the current study was that resistance to the pathogens responsible for fusarium ear rot may see contributions from the microbiota present in the kernels, not being directly/strictly associated to genes of the maize plant alone. The presence of encouraging studies in the literature showing the value of exploring the microbiota of less standardized genotypes to look for bacteria that can protect maize from pathogens [56] further reinforced the hypothesis that this approach could yield interesting results.

The characterization of the embryo-associated bacterial microbiota gave different results depending on the technique employed: cultivation-dependent and -independent approaches did not agree on the composition of the bacterial community. 

The major point on which both approaches converge is that they identified a low number of bacteria from the investigated tissue. This is not surprising as the embryo is a tissue that was considered to be completely sterile for a long time even after the discovery of endophytic communities in other organs of plants and, being subject to very strict regulation and protection by several physical and biochemical barriers, it is very hard to colonize for microbes [57].

While the sequencing-based approach described a bacterial community dominated by Proteobacteria, the cultivation-dependent characterization led to the isolation of mostly Gram-positive bacteria, in particular belonging to the genera *Bacillus* and *Micrococcus*, which is in line with results previously obtained on maize seeds [23]. This could be due to the stress induced by the drying of the seed, a kind of stress that endospore-forming Gram-positive bacteria can endure more easily than other bacteria. With the current data, it is not possible to conclude whether the majority of the bacteria identified through sequencing, in particular the high number of Burkholderiaceae, are not viable and identified only from their remaining DNA, or if they are in a viable but not cultivable state. 

The cultivation-independent approach gave a very low number of reads belonging to bacteria. The sequencing was done three times, and gave poor results when using PNA blockers, and comparable results when using no blocking technique or blocking primers. Our personal experience thus suggests that, when working with maize embryos, the use of PNAs may not be recommended, and the use of blocking primers does not lead to substantially better results compared to the amplification carried out in conventional PCR. 

One experimental trial and two comparisons with previously performed studies can help explain and confirm the validity these results: (i) although the number of reads is below what considered a standard for amplicon sequencing surveys, the results were validated using an independent, quantitative assay: quantification of either total eubacteria or Firmicutes in the samples by digital-PCR. The obtained results validated both the low number of bacteria in the samples and the percentage of Firmicutes compared to the total of eubacteria for most analyzed samples. (ii) The overall structure of the microbiota of the embryos, with a great abundance of Proteobacteria, followed by Actinobacteria, Bacteroidetes, and Firmicutes is in accordance with the results obtained by Guimaraes and colleagues [58] when analyzing the microbiota of maize ears. (iii)The number of reads and OTUs obtained from the sequencing is comparable to those obtained by a previous work carried out on embryos of wheat [57]. 

These three considerations confirm that the low number of bacterial reads and of OTUs is most likely a faithful depiction of what is present inside the embryos.

The analysis of beta-diversity in the microbial communities highlighted that the year of sampling is not a statistically significant variable, while the genotype of the accessions is highly significant, though it explains only a minor part of the variation between the microbial communities. This significant but small variation is in line with previous results that show how the main difference in the selection of bacteria within seeds can be found between wild and domesticated *Zea mays* rather than between landraces and modern hybrids [59]. This variation, despite being small, allows the distinction of two groups between the accessions: one that includes accession N in both years and accession W in 2017, and one that groups all other accessions. The first group is characterized by a higher abundance of Firmicutes, while the second one shows an embryo microbiota dominated by Proteobacteria. It is interesting to note that these accessions have very distinct geographic origins (North Italy and South Africa, respectively), are on completely opposite ends of the spectrum regarding kernel pigmentation (black and white, respectively), and show a different length of their cultural cycle (FAO classification 400 and 800, respectively). The only trait that they show to have in common is higher resistance to fusarium ear rot, as shown by the in-field inoculation assays.

It must be pointed out that the results obtained from the field experiments are related to a different year from any of those in which the microbiota was examined. Accession N had fairly consistent results regarding the microbiota structure in both examined years, so it is sensible to hypothesize that the general microbiota in 2020 was similar to that seen in 2017 and 2018. Accession W, instead, was shown to group in either cluster in the two examined years and, therefore, to conclude whether the microbiota composition is related to its reduced rate of *F. verticillioides* infection through the silk channels, data regarding its microbiota composition in that year would be necessary. Currently, involvement of a genetic resistance factor in accession W, unrelated to microbiota structure, is an equally valid explanation of the obtained results.

While it is tempting to see a cause-effect relationship between the structure of the embryo microbiota and the resistance to fusarium ear rot, other explanations, such as conserved maize genes implicated in fusarium resistance, cannot be discarded. Further studies will be needed to discern the real impact of microbiota composition and host genetics on the resistance/susceptibility to this disease.

Some insights can be taken from the results obtained from the analysis of the cultivable bacteria regarding their ability to exert a biocontrol activity against *F. verticillioides*. A first consideration is that there is no correspondence between the antifungal activity of single strains and the overall resistance to fusarium ear rot of the variety from which the bacteria were isolated. Most of the bacteria with antifungal activity, and those with the highest activity, were isolated from the hybrid accession, which shows severe damage from *F. verticillioides* infection; at the same time, only a few effective strains were isolated from varieties N and W. This can be explained when taking into consideration that (i) the concentration of the bacterial cells for each given strain in the embryo is lower than that used during the experiment, (ii) the exogenous application of the bacteria was carried out on the surface of the seed after sterilization, and (iii) some bacteria may not show antifungal activity on their own but may contribute synergically with other strains to the protection of the seed in natural conditions.

Another consideration is that the biocontrol effect during the in vivo assays was seen only for the bacteria isolated from maize, and not for the bacteria isolated from other plants included in the study. This excludes the possibility that the effect is related to a generic plant response to the high concentration of bacteria to which the seeds were exposed or to a competition for space on the surface of the seed between bacteria and *F. verticillioides*. It is therefore more likely that these strains have co-evolved with the fungal pathogen and have become more effective in restricting the growth of *Fusarium verticillioides* than bacteria that have never encountered this pathogen before. Despite these encouraging preliminary results, the in-field applicability of these strains needs to be demonstrated at various levels: their safety will have to be assessed, as well as their ability to contain the disease in real operative conditions, and their effect on the accumulation of mycotoxins in the kernels. Furthermore, both in the in vivo assay using the bacteria isolated from maize and in the inoculation assay in field, the best results showed an almost complete elimination of the damage caused by the exogenously applied *F. verticillioides* conidia, but seemingly had no effect against the natural infection. This could be explained by a spatial limitation of the assays carried out, that put the bacteria only on the surface of the seed and inoculated the pathogen in maize through the silk channel only, or by different interactions with the *Fusarium* present in field. Further experiments will be needed to understand the precise mechanism behind these interactions.

Should these strains prove effective in field conditions, they could be interesting for their embryo localization which could make them possible to be introduced in the seeds before sowing either by treatment of the flowers [60] or, should they be inheritable, by selective breeding. This would allow to obtain plants that benefit from this protective effect from the moment of germination without needing to apply the bacteria through a coating. Also, the persistence of these strains in the kernels may contribute to a lower accumulation of mycotoxin and/or to a lower spoilage rate, as it has been suggested that the microbiota plays a crucial role in the development of postharvest losses [61].

## 5. Conclusions

This study provided preliminary results regarding the relevance of the embryo-associated bacteria of different maize accessions, both as communities and as single strains, in contributing towards resistance against fungal infections. Further studies are planned to further dissect the importance of these microbe-associated traits and their possible heritability to determine their possible utilization in future breeding programs.

## Figures and Tables

**Figure 1 microorganisms-09-02388-f001:**
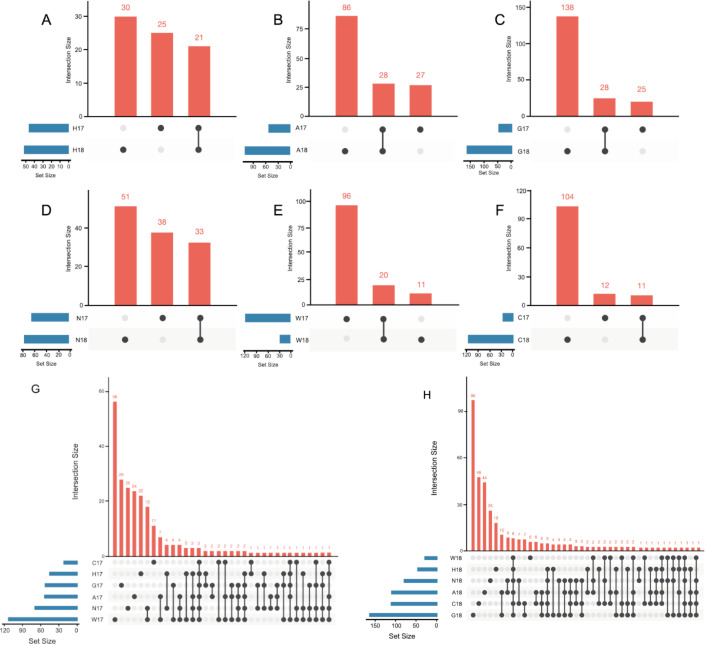
UpSet graphs reporting the unique and shared OTUs within each accession in the two years 2017 and 2018. Each graph reports the result of a different accession: (**A**) accession H; (**B**) accession A; (**C**) accession G; (**D**) accession N; (**E**) accession W; (**F**) accession C. Graphs reporting the comparison of shared and unique OTUs identified between different accessions in (**G**) year 2017 and (**H**) year 2018.

**Figure 2 microorganisms-09-02388-f002:**
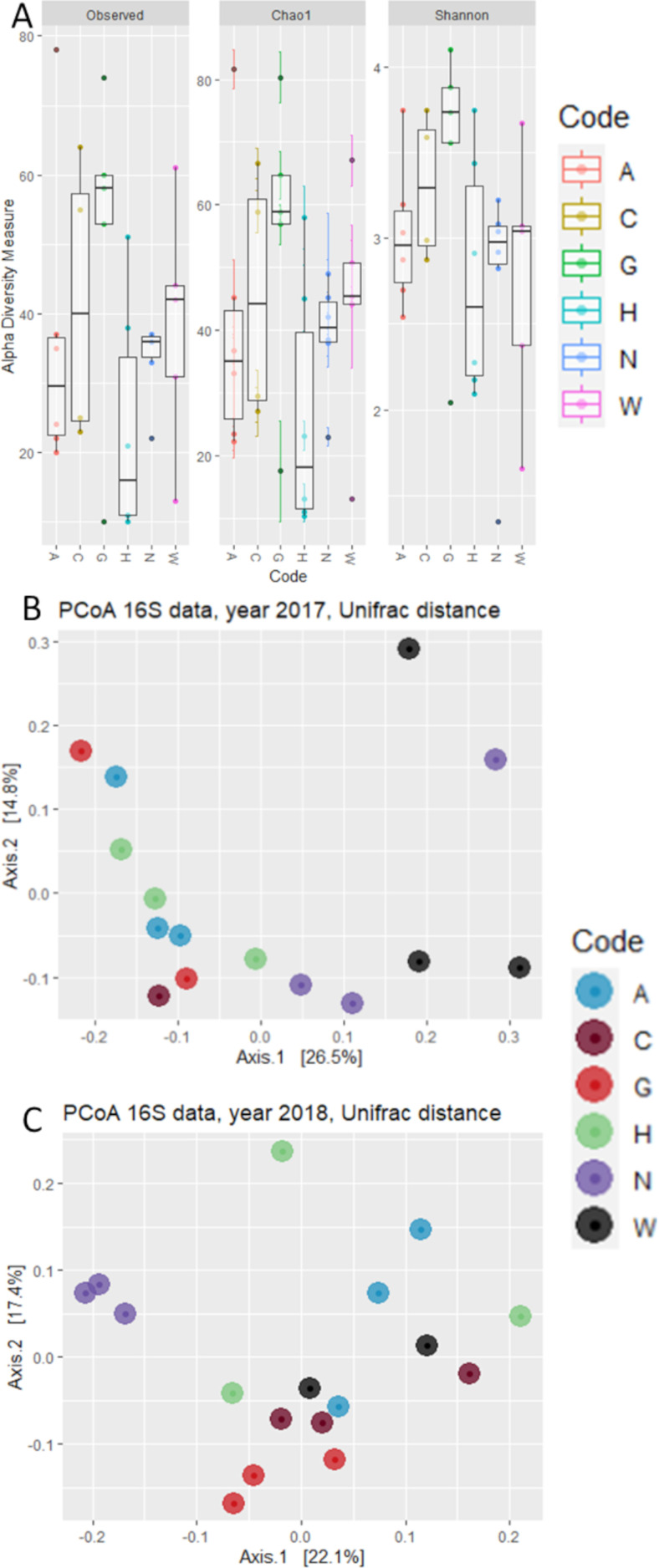
Graphs reporting the (**A**) Alpha and (**B**) Beta diversity of the bacterial communities described by NGS sequencing. (**A**) Box plots reporting the Observed OTUs, Chao-1, and Shannon indexes for each examined accession. (**B**) PCA plot reporting the results of Beta Diversity, calculated with Unifrac algorithm. Shape of the markers indicate year of sampling, while the color indicates the different accessions.

**Figure 3 microorganisms-09-02388-f003:**
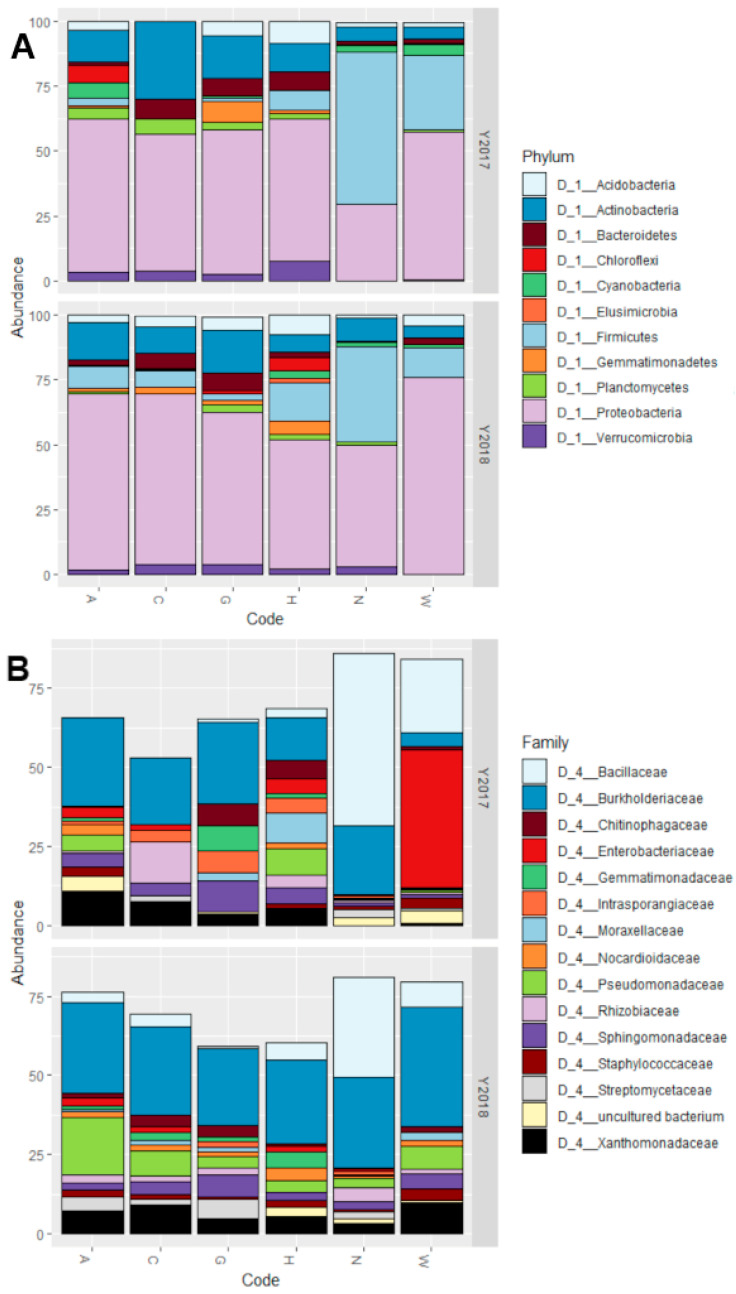
Graphs reporting the microbiota composition as relative abundance (%) at (**A**) phylum level and (**B**) family level. Both graphs show the accession on the X-axis and the percentage of relative abundance on the Y-axis. The Y-axis is also divided in two grids, one for year 2017 and one for year 2018. Bars reaching a total below 100% in (**B**) are due to OTUs assigned to families other than the most abundant 15 not being plotted on the graph.

**Figure 4 microorganisms-09-02388-f004:**
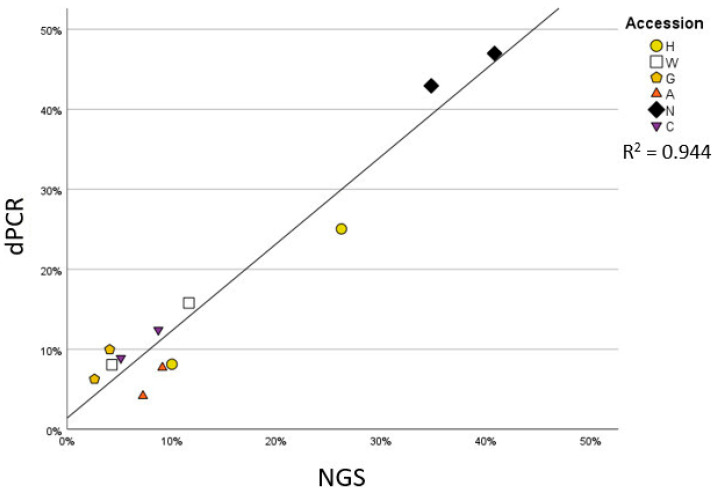
Scatter plot reporting the percentage of Firmicutes on all bacteria obtained by two different methods: the Y-axis reports the results of quantification with digital PCR, reporting the percentage of the target copy number identified when using specific primers for Firmicutes compared to a general eubacteria primer pair; the X-axis reports the percentage of reads belonging to Firmicutes compared to total bacterial reads obtained by NGS sequencing. The graph also reports the trendline and R2 value.

**Figure 5 microorganisms-09-02388-f005:**
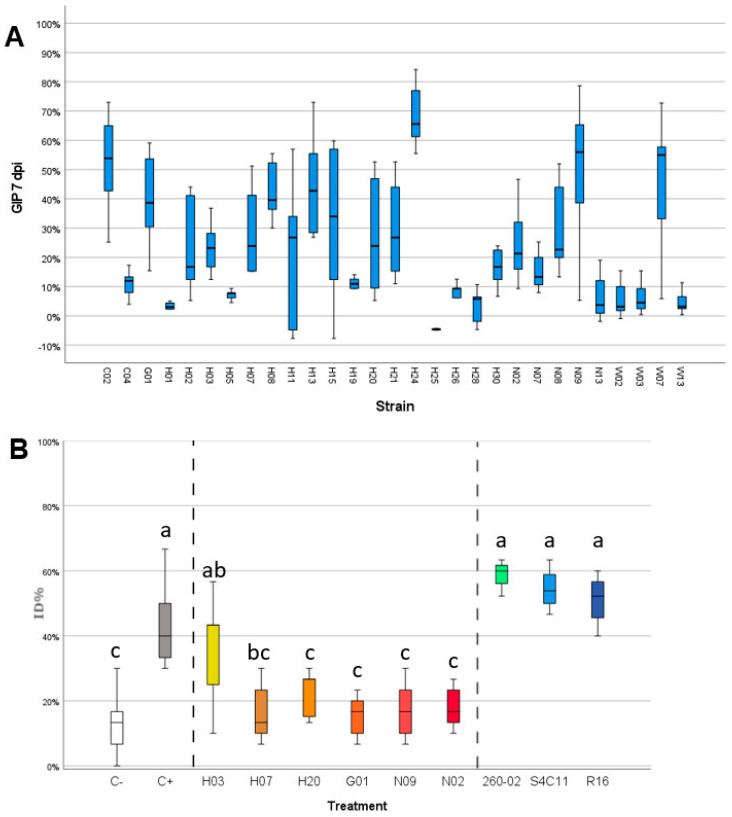
Graphs reporting the results of the in vitro (on agar in Petri dishes) antifungal assay and in vivo (on germinating kernels in Petri dishes) biocontrol assay against FV and FVm, respectively. (**A**) Graph reporting on the X-axis the different bacterial isolates and on the Y-axis the growth inhibition percentage obtained at 7 dpi. (**B**) Graph reporting on the X-axis the different treatments carried out on maize and on the Y-axis severity of FV-induced symptoms, expressed as infection percentage index (ID%). Vertical dashed lines separate between groups of samples: negative (C−) and positive (C+) controls, bacteria isolated from maize in this study (H03, H07, H20, G01, N09, N02), and bacteria isolated from other hosts in previous studies (260-02, S4C11, R16). Different letters (a, b, c) indicate statistically significant differences among results, according to a one-way ANOVA analysis followed by Tukey’s post-hoc test (*p* = 0.000).

**Figure 6 microorganisms-09-02388-f006:**
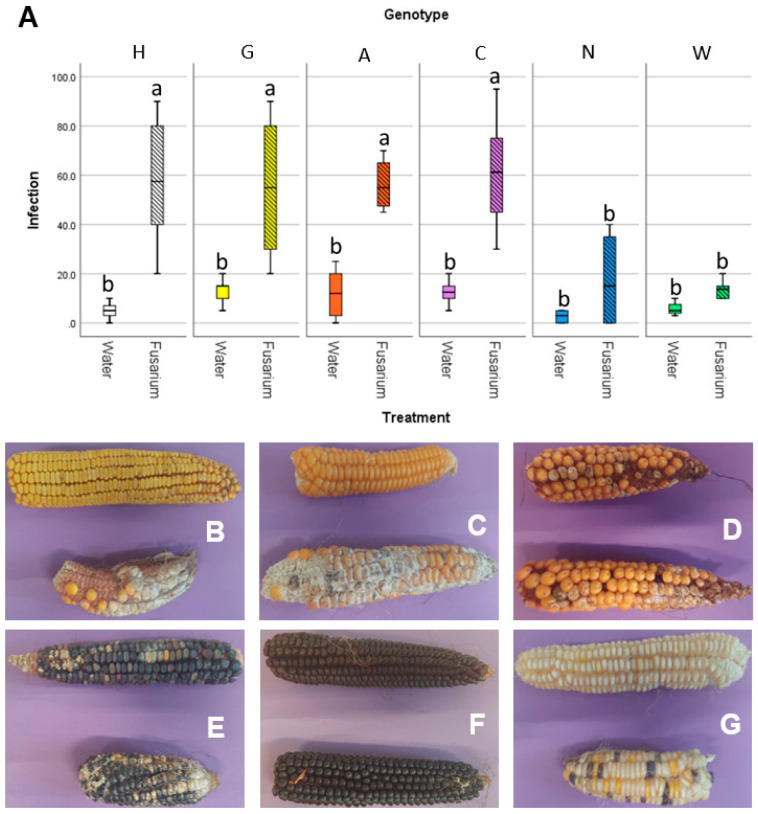
Results of the in field inoculation with FVm on maize ears. (**A**) graph reporting on the X-axis the different accessions, either mock-inoculated with water or inoculated with FVm, and on the Y-axis the infection severity, expressed as a percentage of the ear surface showing symptoms. Different letters (a, b) indicate statistically significant differences among results, according to a one-way ANOVA analysis followed by Tukey’s post-hoc test (*p* = 0.012). Panel showing representative maize ears either mock inoculated (top of each picture) or FVm-inoculated (bottom of each picture) for (**B**) accession H, (**C**) accession G, (**D**) accession A, (**E**) accession C, (**F**) accession N, and (**G**) accession W.

**Table 1 microorganisms-09-02388-t001:** List of maize accessions used in the study.

Accession	Code	Genetic Constitution	Origin	FAO Classification	Pigmentation	Reference
B73 × Mo17	H	Hybrid	USA	700	Yellow	[25]
Ottofile Basia	A	opv	North Italy	300	Orange (Phlobaphenes)	Unpublished results of our group
Ottofile Tortonese	G	opv	North Italy	300	Yellow	[25]
Spinoso Nero della Val Camonica	N	opv	North Italy	400	Black (Phlobaphenes)	[16]
Mais Bianco Qwa-Qwa	W	opv	South Africa	800	White	[26]
Millo Corvo	C	opv	Spain	400	Black (Anthocyanins)	[15]

**Table 2 microorganisms-09-02388-t002:** Table reporting the number of bacterial isolates obtained from each studied maize accession. The table reports the initial number of isolates, the number after each screening step, and the number of different bacterial genera among them.

Accession Code	Isolated Colonies	After Morphology Screening	After RAPD	N° of Genera
H	50	31	30	7
A	28	6	5	5
G	82	3	2	2
N	18	17	13	4
W	62	21	16	2
C	63	4	4	2

**Table 3 microorganisms-09-02388-t003:** Table reporting the average value of total bacterial reads and assigned OTUs after NGS analysis of each accession in both years.

Accession Code	Year	Reads	OTUs
H	2017	548	46
	2018	560	51
A	2017	312	55
	2018	674	114
G	2017	162	53
	2018	694	166
N	2017	836	71
	2018	371	84
W	2017	1163	116
	2018	112	31
C	2017	53	23
	2018	522	115

## Data Availability

Data presented in this study can be found as part of the manuscript, and in the Appendix A. Other information can be retrieved from the following sources: sequences of the 16S of each bacterial isolate is available on NCBI at Accession Numbers from OK584315 to OK584384 and reported isolate-by-isolate in Appendix A; the sequencing reads utilized in this analysis are available at ENA PRJEB47936; the files and script–OTU table, mapping files, R script–used in this study are available on GitHub (https://github.com/AlessandroPasser/Maize_Embryo_Microbiota (accessed on 16 November 2021)).

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
