# Peer review of "Bacterial Communities in the Embryo of Maize Landraces: Relation with Susceptibility to Fusarium Ear Rot"

_microorganisms, 2021, doi:10.3390/microorganisms9112388_

Round 1
Reviewer 1 Report
A very interesting study and well written up. Its a pity that your in vitro bioassay results didn't line up well with your in vivo results, but that is one of the major difficulties that all biologists face. Although you attempted to culture as many microbes as possible, you never know how many you might have missed or which were the most significant at which point in the infection cycle - why did you limit the sampling to embryos only? I've made extensive comments in the attached PDF asking for minor changes to figures, grammar, extra details in the methods, further discussion on certain points (in particular more about the biology of fusarium ear rot infections) and inclusion of extra references. I look forward to reviewing your changes.

Author Response
We thank the reviewer for their very accurate, precise, and in-depth evaluation of our manuscript.
The various corrections to form, grammar, and spelling were implemented and can be found in the new version of the manuscript. For more elaborate comments, here is a point-by-point rebuttal detailing the corrections carried out. Please note that the reference to line numbers are referring to the lines in the manuscript with Track Changes.
Comment 1: Line 42: double check this date? Maybe put the actual year it came to Europe in brackets to disambiguate
Reply: We revised with a more precise temporal indication of the arrival of maize in Europe. [Line 42]
Comment 2: Line 96: if you want to back this up with a recent reference, check out this publication: Seed-Transmitted Bacteria and Fungi Dominate Juvenile Plant Microbiomes
Reply: We thank the reviewer for this suggestion and included the article as a reference. [Line 85]
Comment 3: Lines 101-106: You say seed here, but you actually limited yourself to embryo only. Would you please specify that here and explain why? For example, you might have included the endosperm in your sampling, which may have resulted in additional endophytes (but you didn't).
Reply: We corrected the reference to seed-associated microbiota to being embryo-associated microbiota. As for the choice of using only the embryos, a first procedure in our study – not discussed in the manuscript in its current version – was aimed at making this choice: we carried out the same isolation procedure on ten samples of whole seeds, embryos only, and endosperm only. From the samples of endosperm only we obtained very few bacterial colonies, on only 3 out of 30 plates, and instead obtained very extended fungal contaminations. The samples of whole seeds showed bacterial colonies on almost all the plates, but still had problems with the growth of fungi – with a Fusarium-like appearance – that made isolation of the bacterial colonies difficult. The samples coming from embryo only showed a comparable number of bacterial colonies to the whole seed ones but only a fraction of the fungal contaminants. Therefore, we decided to use the embryo-only approach in our whole experiment. We added this information in materials and methods [Lines 126-128] and with an additional piece of supplementary information, the new Table S1.
Comment 4: Line 152: Include reference in the bibliography.
Reply: We added the reference to the method to the bibliography. Also, in accordance with the comments from another reviewer, we expanded this section to give a brief overview of the method.
Comment 5: Lines 155-157: please include at least one sentence describing this method (ie. generating a fingerprint of differently sized DNA amplicons)
Reply: We added a sentence giving brief details about the method. [Lines 162-163]
Comment 6: You should try doing agglomerative heirarchical clustering with Bray-Curtiss Dissimilarity or PCA on the same data after transformation into presence and absence (0 or 1). When you have wonky sequencing data perhaps caused by dirty DNA or saturated PCR, abundance data becomes very noisy and perhaps meaningless. Maybe consider analyzing the two years of data separately.
Reply: We thank the reviewer for this suggestion. We carried out analyses also splitting the years 2017 and 2018, but analyzing the data in that way removed the ability to confirm through ADONIS the significance of the year variable and of the year*genotype effect. Therefore, we decided to keep the current presentation of data with both years compared in the same analysis.
Also, we carried out analysis of beta-diversity using Unweighted Unifrac, which is sensitive to unique taxa in the same way as the 0/1 approach suggested by the reviewer. The results obtained from this approach were slightly different from those using Weighted Unifrac, but did not change the statistical significance of the variables. The results of the ADONIS analysis on the Weighted Unifrac were added in the manuscript to show that the considerations regarding the statistical significance of genotype on the composition of microbiota are solid also when disregarding abundance data. [Lines 432-436].
Ultimately, even if we present the results of both Weighted and Unweighted Unifrac distances through ADONIS, we decided to retain the weighted unifrac plots as a main figure since the digital PCR confirmed the accuracy of quantification.
Comment 7: Line 280-281: what criteria was used to decide what was "potentially harmful"? How many strains were eliminated from the experiment because of this?
Reply: The criteria used were that these bacteria belonging to species reported in literature as pathogens or food contaminants. We added this information in the sentence [Lines 302-303]. The number of strains that were removed from this taxonomical screening is already reported in the Results section [Lines 369-371]
Comment 8: Lines 355-360: If you got the best results without PNA Blockers, say that here and explain what method was used on the samples you're reporting here. How many total reads were obtained and how many were discarded for being low quality, chloroplast or mitochondria? A single run on an Illumina Miseq machine should yield millions of reads and doing similar sequencing of bacteria in seeds I've got as little as 72 reads and as many as 37590 reads before.
Reply: We moved the numbers regarding the other sequencing methods that were tried (no blocking and PNA blockers) from the Discussion section to here [Lines 379-387]. Also, we added the total number of reads obtained, before the filtering for chloroplasts, mitochondria, and quality filtering. [388-391]
Comment 9: Figure 1: Need higher resolution of this figure. Would also be helpful if you labelled each accession within its corresponding chart so that the reader doesn't need to refer to the caption. For A-F, keep the order of the X axis the same so 17 is first and on the top, then 18 next which is on the bottom, then both comes last.
Reply: We completely remade the figure to obtain a higher-quality version. In the process, we corrected the placement of the bars so that the data relating to year 2017 always came before those of year 2018.
Comment 10: Figure 2B: these shouldn't be black or they look like W. Maybe fill them with white, with a black outline
Reply: We corrected this aspect of the figure, making the markers in the legend empty to avoid confusion with the W accession.
Comment 11: Figure 3B: Why is there this unreported space here? If this represents unknown or "other families" please indicate as such.
Comment 12: Lines 418-422: You put the wrong caption here (its from Figure 2), please put in caption 4
Reply: The unreported space is indeed “other families”. The detail was lost in the problem with the figure caption. We corrected the figure caption, which now reports what the unreported space is. [Lines 457-460]
Comment 13: Line 463: shouldn't this be ID%?
Reply: We corrected the abbreviation. [Lines 504, 515]
Comment 14: Line 470: you don't know which endogenous microbes might be rotting the seed and its unlikely to only be Fusarium; better to just say "endogenous microbes"
Reply: We did not carry out an in-depth characterization of the pathogenic microbes present in the seeds, but the symptoms that were caused matched perfectly the typical ones caused by Fusarium infection. We corrected the sentence to express that, as suggested by the reviewer, other microorganisms might be involved in the process, but still inform the reader that the visual assessment of symptoms points towards Fusarium being involved. [Lines 504-506, 512-513]
Comment 15: Figure 5A: This graph should include positive and negative controls as well. How do you explain the negative value for H25?
Reply: The graph in question does not have positive and negative controls to include: the GIP, being calculated as a percentage of growth reduction compared to the negative control (growing without any bacteria), cannot be sensibly calculated for the negative control, as it would mean comparing the negative control with itself and would always give exactly 0% of GIP; positive controls, for example including fungicides, were not used in the experiment.
Regarding the negative value of H25, it could be explained with the bacterial strain producing metabolites that caused a positive response in F. verticillioides, stimulating its growth, rather than inhibiting it. The presence and nature of such metabolites was not assessed, but it is the conclusion to which we came.
Comment 16: Figure 5B: consider putting some illustrative photos here of strong inhibition on a agar and no inhibition on agar. Also strong Fusarium infection on germinating seeds and inhbited infection on germinating seeds.
Reply: We added visual examples of the different results obtained in the in vitro and in vivo inhibition assays as an additional supplementary figure in the manuscript, a new Figure S1. The figure is referenced in the materials and methods in sections 2.3.1 and 2.3.2. The previous Figure S1 that showed the results of the RAPD analysis was renamed as Figure S2.
Comment 17: Discussion section: Please discuss a little more in depth about the biology of fusarium ear rot in the field. Does it only go in through the silks and rot the seed from the embryo-out, or can it also come in through pollen, breaks in the seed coat (rotting from outside-in), vectoring by insect feeding, other phenomenon? Perhaps there are different types of fusarium ear rot infection and certain types of biocontrol might work to control different types? What about other types of fungal ear rots not caused by Fusarium; some of the natural endogenous infections may have been caused by these other fungi (ie. Diplodia spp., Penicillium spp., Aspergillus spp., Gibberella spp.)? Read this paper and reference it if useful: Fungal Pathogens of Maize Gaining Free Passage Along the Silk Road
Reply: All the methods of infection mentioned by the reviewer can contribute to infection of maize seeds in field. For Fusarium the two most important methods of infection are entrance through the silk channels and using holes caused by insect feeding, in particular O. nubilalis. In our experimental field, infections by other pathogens are present too, in particular Aspergillus and Ustilago, but are less relevant in incidence and damage caused.
In accordance with the reviewer’s suggestion, we added more information regarding these additional infection routes and other pathogens that, while not directly investigated by our study, are useful to understand the dynamics of what happens in the maize field. [Lines 563-572]
Comment 18: Lines 527-529: Could you please mention other studies which have attempted similar experiments and give you reason to believe your approach might work (please reference)? This paper seems to have tried to do very similar things and had good results in planta: Bacterial endophytes from wild maize suppress Fusarium graminearum in modern maize and inhibit mycotoxin accumulation.
Reply: We thank the reviewer for the suggestion of this article. It was included in the discussion. [Lines 578-581]
Comment 19: Lines 553-556: is there any explanation for this result? How did PNA blockers affect the diversity detected?
Reply: We do not have an explanation for why the results obtained with PNAs were worse than those obtained without them. The same quality of results could be explained by the high concentration of organelles in the embryo cells, making the sheer quantity of organellar DNA so high that the use of PNAs is not enough to significantly reduce the frequency of mitochondrial and plastidial reads obtained. This is in line with the results obtained with the blocking primers which also yielded more than 99% of organellar sequences when used on embryos, unlike what is reported in literature regarding their efficacy.
As for the diversity detected, the run with PNA yielded overall less diversity, as all bacterial sequences obtained belonged to Actinobacteria, Bacteroidetes, Firmicutes, Planctomycetes, or Proteobacteria, failing to detect bacteria belonging to 6 other Phyla, which were identified in the final experiment. Whether this is caused by some specific bias towards these Phyla, or just given by the low number of reads obtained which made members of less abundant Phyla impossible to detect, we cannot say for certain.
Comment 20: Lines 576-580: Clumsy/confusing, please rewrite for clarity.
Reply: We rephrased the sentence and split it into two. The new version reads as follows “This variation, despite being small, allows the distinction of two groups between the accessions: one that includes accession N in both years and accession W in 2017, and one that groups all other accessions. The first group is characterized by a higher abundance of Firmicutes, while the second one shows an embryo microbiota dominated by Proteobacteria.” [Lines 635-641].
Comment 21: Lines 604-606: Another important consideration is that the endophytes responsible for the effect observed in the field, were in fact not isolated and cultured. Furthermore, while N evidently had a high level of Firmicutes, W did not, which suggests they may possess different mechanisms of resistance (ie. N has endophytes protecting it, W has genetic resistance). Please discuss.
Reply: We modified the discussion to include the considerations suggested by the reviewer, ad also a comment from Reviewer 2, mitigating the statements regarding accession W. [Lines 647-656]
Comment 22: Lines 637-639: you should also include here OTU count tables. Does tables S1 include 16S sequences?
Comment 23: Lines 651-656: Please repeat what data is supplied in the supplimental materials. Also write this here: The sequencing reads utilized in this analysis are available available at ENA PRJEB47936, while the files and script are available on GitHub (https://github.com/AlessandroPasser/Maize_Embryo_Microbiota)
Reply: Table S2 (previously Table S1) includes the accession number to retrieve from NCBI each single 16S sequence obtained with Sanger sequencing, along with other data. The OTU count tables are instead provided on the GitHub link. Also, we’re sorry to have overlooked the fact that the Data Availability Statement was just a placeholder in the submitted version of the manuscript. The correct Data Availability Statement can be found in the new version of the manuscript. [Lines 725-730]
Reviewer 2 Report
The manuscript is well written. However, I have some questions.
- I was wondered that the selection of strains from the maize embryo to verify the in vivo biocontrol assay against FVm. As we could see in Table_S1, I thought the antifungal activity of C02 was higher than G01 chosen in this paper, and the same situation was found between the W07 and N09. The reason was described as “the taxonomic attribution (ignoring strains that belong to species known as possibly pathogenic), the GIP produced, and their ability to grow rapidly and consistently in laboratory.”(Line 451-452). From my perspective, the strains C02 and W07 were better to continue the next experiment.
- We could easily know that the diversity of the bacterial communities described by NGS sequencing was certainly different in 2017 and 2018. But the result of 3.4 was based on the field trial of 2020. I suppose the field experimental result was indirectly obtained.
- All the figures in this paper were not very clear. Hope you can change them to the higher resolution version.
- Apparently, a few format errors are in the manuscript, for example, line 462 and 472, I%I, and the other places also have some errors, which need to be carefully reviewed and modified.
Author Response
We thank the reviewer for their careful evaluation of our manuscript and for their comments and questions.
Please note that the reference to line numbers are referring to the lines in the manuscript with Track Changes.
Comment 1: I was wondered that the selection of strains from the maize embryo to verify the in vivo biocontrol assay against FVm. As we could see in Table_S1, I thought the antifungal activity of C02 was higher than G01 chosen in this paper, and the same situation was found between the W07 and N09. The reason was described as “the taxonomic attribution (ignoring strains that belong to species known as possibly pathogenic), the GIP produced, and their ability to grow rapidly and consistently in laboratory.”(Line 451-452). From my perspective, the strains C02 and W07 were better to continue the next experiment.
Reply: As the reviewer points out correctly, strains C02 and W07 gave higher GIP than G01 and N09. They were not selected for the trials because of the last of the points used to identify the strains that should be used in further experiments: “their ability to grow rapidly and consistently in laboratory”. Strains C02 and W07 were quite slow in growth and had troubles in growing in liquid medium, only rarely reaching acceptable levels of turbidity that would allow the preparation of the samples for the in vivo assay. For this reason, they were not considered for these assays.
Comment 2: We could easily know that the diversity of the bacterial communities described by NGS sequencing was certainly different in 2017 and 2018. But the result of 3.4 was based on the field trial of 2020. I suppose the field experimental result was indirectly obtained.
Reply: Seeing the difference in NGS results between years 2017 and 2018, it is quite likely that the microbiota in the embryos in 2020, when susceptibility was assessed in field, was different, especially for accession W, which showed more variation among the two years examined. We added a sentence to express this concern in the Discussion. “It must be pointed out that the results obtained from the field experiments are related to a different year from any of those in which the microbiota was examined. Accession N had fairly consistent results regarding the microbiota structure in both examined years, so it is sensible to hypothesize that the general microbiota in 2020 was similar to that seen in 2017 and 2018. Accession W, instead, was shown to group in either cluster in the two examined years and, therefore, to conclude whether the microbiota composition is related to its reduced rate of F. verticillioides infection through the silk channels, data regarding its microbiota composition in that year would be necessary. Currently, involvement of a genetic resistance factor in accession W, unrelated to microbiota structure, is an equally valid explanation of the obtained results.” [Lines 647-656]
Comment 3: All the figures in this paper were not very clear. Hope you can change them to the higher resolution version.
Reply: We tried to improve the quality of the pictures, in particular Figure 1, also operating some changes taking into consideration comments on the figures from other reviewers. We hope that the new versions are satisfactory.
Comment 4: Apparently, a few format errors are in the manuscript, for example, line 462 and 472, I%I, and the other places also have some errors, which need to be carefully reviewed and modified.
Reply: We thank the reviewer for having pointed out these mistakes and proceeded to fix them.
Reviewer 3 Report
The manuscript title “Bacterial Communities in the Embryo of Maize Landraces: Relation with Susceptibility to Fusarium Ear Rot” discusses the bacterial community existence in five different maize landraces and their interactive role in Fusarium Ear Rot. The authors used the 16S rRNA gene analysis in the study and In vitro characterization of the antifungal properties of the isolated bacteria. The author concluded that Firmicutes are the important bacterial taxa.
The author needs to fix and justify the few issues in the manuscript.
- Line 152-174: The author should add a brief protocol of DNA isolation, and add the details about the reaction mixture of PCR and how the product amplifies.
- Line 178: Add the DNA isolation protocol.
- Figure 3: I think the author used abundance percent in the graph, and Figure 3B need to revise as the abundance distribution should be 100%.
- Figure 4: dPCR had a red line, need to remove.
Author Response
We thank the reviewer for their comments on the manuscript and for their evaluation of our study.
Please note that the reference to line numbers are referring to the lines in the manuscript with Track Changes.
Comment 1: Line 152-174: The author should add a brief protocol of DNA isolation, and add the details about the reaction mixture of PCR and how the product amplifies.
Reply: We added more details in materials and methods as requested. The new information can be found in Lines 154-157 and 173-180.
Comment 2: Line 178: Add the DNA isolation protocol.
Reply: We added more details in materials and methods as requested. The new information can be found in Lines 191-194.
Comment 3: Figure 3: I think the author used abundance percent in the graph, and Figure 3B need to revise as the abundance distribution should be 100%.
Reply: Figure 3 describes abundance as percentage, the missing parts to 100% in Figure 3B are from families of low abundance that are not plotted in the graph. As a comment from another reviewer pointed out to us, the previous caption of Figure 3 was incorrect. The new caption is correct and explains this unreported percentage. [Lines 457-460].
Comment 4: Figure 4: dPCR had a red line, need to remove.
Reply: We have corrected the figure as suggested.